# Comparable Studies on Nanoscale Antibacterial Polymer Coatings Based on Different Coating Procedures

**DOI:** 10.3390/nano12040614

**Published:** 2022-02-11

**Authors:** Thorsten Laube, Jürgen Weisser, Svea Sachse, Thomas Seemann, Ralf Wyrwa, Matthias Schnabelrauch

**Affiliations:** 1INNOVENT e.V., Biomaterials Department, 07745 Jena, Germany; jw1@innovent-jena.de (J.W.); ss5@innovent-jena.de (S.S.); rw1@innovent-jena.de (R.W.); ms@innovent-jena.de (M.S.); 2INNOVENT e.V., Surface Technology Department, 07745 Jena, Germany; ts2@innovent-jena.de

**Keywords:** quaternary ammonium compounds, phosphonate, atom transfer radicalpolymerization, antibacterial activity, nanoscale, coating

## Abstract

The antibacterial activity of different antibiotic and metal-free thin polymer coatings was investigated. The films comprised quaternary ammonium compounds (QAC) based on a vinyl benzyl chloride (VBC) building block. Two monomeric QAC of different alkyl chain lengths were prepared, and then polymerized by two different polymerization processes to apply them onto Ti surfaces. At first, the polymeric layer was generated directly on the surface by atom transfer radical polymerization (ATRP). For comparison purposes, in a classical route a copolymerization of the QAC-containing monomers with a metal adhesion mediating phosphonate (VBPOH) monomers was carried out and the Ti surfaces were coated via drop coating. The different coatings were characterized by X-ray photoelectron spectroscopy (XPS) illustrating a thickness in the nanomolecular range. The cytocompatibility in vitro was confirmed by both live/dead and WST-1 assay. The antimicrobial activity was evaluated by two different assays (CFU and BTG, resp.,), showing for both coating processes similar results to kill bacteria on contact. These antibacterial coatings present a simple method to protect metallic devices against microbial contamination.

## 1. Introduction

Bacteria and microbial pathogens, which cause serious infections and diseases, have been a threat to both, human health and social development [1]. Outbreaks of infectious diseases caused by bacteria, viruses, and fungi are among the leading challenges of the health systems worldwide, responsible for more than a quarter of global deaths annually [2,3,4,5,6]. The control and prevention of microbial infections, especially in healthcare or public facilities, is a tremendous challenge because microorganisms are omnipresent and can be spread through air, water, and all kind of surfaces [7]. To combat these microbial pathogens, all kinds of antimicrobial agents, including antibiotics, disinfectants, antiseptics, and nano particles, have been developed [8,9,10,11,12,13,14]. However, the widespread and careless use of antibiotics and disinfectants has led to the appearance of new strains of antimicrobial-resistant microorganisms, dramatically exacerbating the antimicrobial problem [15,16,17,18,19,20,21,22]. Many microbial pathogens acquire resistance to conventional antimicrobial agents based on natural or low molecular weight compounds. Furthermore, antimicrobial agents can lead to environmental contamination and toxicity to the human body due to biocidal diffusion [23,24]. In contrast to low molecular weight compounds, antimicrobial polymers show better antimicrobial efficacy and lower residual toxicity due to higher stability and non-volatility, and thus having a long-term activity [25,26,27,28,29].

Antimicrobial polymers can be roughly classified into two kinds of materials: leaching [30,31,32] and non-leaching, contact-active materials. While leaching polymers consist of a polymeric matrix that is used as a carrier material for the active ingredient (e.g., antibiotics, metal ions), non-leaching polymer systems consists of polymers, which are the antimicrobial active ingredient themselves, being cationic and contact-killing. Leaching polymer surfaces are quite popular because they can produce a high concentration of the antimicrobial agents locally. Their disadvantage is that they fail once the leaching component is exhausted. In non-leaching polymers, it is indispensable for an antibacterial effect that the bacteria come very close to the surface, but their effect is not exhausted. The field of such contact-killing antimicrobial polymer surfaces has been extensively reviewed [33,34,35,36,37,38]. Among the antimicrobial polymers (e.g., halamines, poly lysine, poly guanidine, quaternary nitrogen groups, antimicrobial peptides [39]), quaternary ammonium compounds (QAC) are probably the most widely used and studied compounds [40,41]. Quaternary ammonium compounds (QACs) are of particular interest in the control of bacterial infections due to their variability. They belong to the group of cationic surfactants, which are already used for application in various fields, e.g., medicine, food, and textile industry or in cosmetic products [42,43,44,45,46]. QACs may consist of a positively charged nitrogen atom incorporated into an aromatic ring (e.g., pyridinium, imidazolium, quinolinium, and isoquinolinium) or which is covalently bonded to four carbon atoms as a non-cyclic structure (e.g., benzalkonium or cetrimonium), with at least one substituent being an n-alkyl chain of different carbon lengths (C6-C18) [43,47,48,49,50,51,52,53]. It is accepted that QACs are adsorbed to the negatively charged cell surface by the positively charged nitrogen atoms via electrostatic interaction, and then the long lipophilic alkyl chain promotes diffusion through the cell wall. This destroys the cytoplasmic membrane and causes a loss of cytoplasmic components, leading to the death of the cell [54,55,56,57]. QACs show activity at very low concentrations against a broad spectrum of pathogens with e.g., bacteriostatic, sporistatic, tuberculostatic, algistatic activity [58]. Furthermore, in higher concentrations they exhibit cidal activities against bacteria, yeast, and enveloped viruses [59]. Gram-positive bacteria are generally more sensitive to QACs compared to Gram-negative bacteria, mainly due to differences in bacterial cell wall structure and difficulty in penetrating the outer membrane layer of Gram-negative bacteria [60], in which the length of the alkyl chain plays a significant role [61]. The maximum activity of these compounds against Gram-positive bacteria was observed at a carbon chain length of *n* = 12–16, while for Gram-negative bacteria the maximum activity was achieved at an alkyl length of *n* = 14–18. Compounds with chain lengths n ≤ 4 and n ≥ 18 are mostly inactive [46].

Atom transfer radical polymerization (ATRP) is a special form of living/controlled free radical polymerization (LFRP), which was first described in 1995 [62,63,64,65,66,67]. ATRP has become one of the most important synthesis techniques in the polymer industry since its discovery, because it does not require stringent experimental conditions and allows controlled polymerization and block copolymerization of a wide range of functional monomers, resulting in polymers with a narrow molar weight distribution [68] and a desired monomer composition [69] as well as a molecular architecture [70,71]. In addition, ATRP is tolerant to monomers with polar functionality, allowing direct polymerization of functional monomers without the need for tedious protection and deprotection procedures and also polymerization on surfaces can be carried out [72,73]. Initially, polymer systems containing quaternary ammonium groups were prepared and tested for their antibacterial activity [74,75].

In this study, we focused on two different antibacterial polymer systems. On one hand, we chose atom transfer radical polymerization (ATRP), a surface polymerization approach leading directly from the monomers to anti-microbially coated titanium surfaces. For comparison purposes, we used a classical route comprising co-polymerization of quaternary ammonium group-containing vinyl benzyl monomeric building blocks with p-vinyl benzyl phosphonate as adhering monomer [76,77]. In both approaches starting from p-vinyl benzyl chloride, two quaternary ammonium salts were prepared with two dimethyl alkylamines of different alkyl length: octylamine and octadecylamine leading to vinyl benzyl dimethyl octyl ammonium chloride (VBCOQ) and vinyl benzyl dimethyl octadecyl ammonium chloride (VBCODQ), respectively. In this classical route, the antimicrobial coatings were produced by drop coating the metallic surfaces and the phosphonate monomeric moiety was responsible to attach to the titanium surface. For the drop coating variants, these monomers were polymerized together with a phosphonate [76,77] responsible for binding the co-polymer on Ti surface. The two different coating procedures should lead to two different coating structures on the surface (see Figure 1). The coatings produced in different ways were characterized with regard to their coating elemental composition and layer thickness using X-ray photoelectron spectroscopy (XPS). Furthermore, the cytocompatibility of the coating were studied by means of life/dead and WST-1 assays and the antibacterial activity was studied performing BGT and CFU assays. The aim was to find differences in the coating strategies in view of thickness or the antibacterial activity.

## 2. Materials and Methods

### 2.1. Materials

*p*-Vinyl benzyl chloride (VBC) and AIBN (azo-bis-(isobutylonitril)) were purchased from Sigma-Aldrich (Taufkirchen, Germany). 4-(Chloromethyl)phenyltrichlorosilane was purchased from ABCR (Karlsruhe, Germany). PMDETA (N,N,N′,N″,N″-pentamethyldiethylentriamin) was purchased from Acros (Geel, Belgium). NaI, NaH, Cu(I)Cl, Cu(II)Cl_2_, triethyl amine, ethyl acetate, acetonitrile, acetone, ethanol, THF, DMF, toluene, diethyl ether, pentane, dichloromethane, methanol and chloroform were purchased from Fisher Scientific (Schwerte, Germany), dimethyl octylamine from TCI (Eschborn, Germany). All chemicals and solvents are used without further purification. Si wavers were purchased from Si-Mat-Silicon Materials e.K. (Kaufering, Germany).

### 2.2. Monomer Synthesis

#### 2.2.1. Diethyl *p*-Vinyl Benzyl Phosphonate (VBP)

The synthesis was performed according to [78]. NaH (60% in grease; 6.65 g; 163 mmol) was suspended in 150 mL THF at 0 °C. Diethylphosphite (28.8 g; 163 mmol) was added dropwise. The solution was allowed to come up to room temperature, transferred in a dropping funnel, and slowly added to a solution of VBC (27.64 g; 163 mmol) and NaI (2.44 g; 163 mmol) in 150 mL THF at 0 °C. The solution was stirred overnight, allowing to come up to room temperature. The same amount of ethylacetate was added and the solution was filtered over celite. After removing the solvents in vacuum, further purification was carried out through flash chromatography (ethyl acetate) to yield diethyl *p*-vinyl benzyl phosphonate (39 g; 94%) as a pale-yellow oil. ^1^H NMR (CDCl_3_): 7.36 (2H), 7.26 (2H), 6.69 (dd, 1H, J = 10.8 Hz, J = 17.5 Hz), 5.73 (dt, 1H, J = 17.5 Hz, J = 1 Hz), 5.23 (dt, 1H, J = 17.5 Hz, J = 1 Hz), 4.01 (4H), 3.14 (d, 2H, J = 21.8 Hz), 1.25 (6H). ^13^C NMR (CDCl_3_): 136.22, 135.99; 130.98, 129.74, 126.17, 113.46, 61.91, 33.86, 32.76, 16.15. ^31^P NMR (CDCl_3_): 27.3. IR: *v* = 2981, 1631, 1512, 1409, 1245, 1163, 1051, 1020, 984, 725 cm^−1^.

#### 2.2.2. *p*-Vinyl Benzyl Phosphonic Acid (VBPOH)

The synthesis was carried out according to [78]. Diethyl *p*-vinyl benzyl phosphonate (10 g, 39.4 mmol) was dissolved in dichlormethane (50 mL), bromotrimethylsilane (18.08 g, 190 mmol) was added and the reaction was stirred overnight. The solvent was removed in vacuum and the residue was hydrolyzed with water. The white solid was filtered and recrystallized from acetonitrile to give white needles (4.3 g, 55%). ^1^H NMR (DMSO-D6): 10.51 (2H), 7.37 (d, 2H, J = 7.92 Hz), 7.23 (dd, 2H, J = 8.14 Hz, J = 2.05 Hz), 6.70 (dd, 1H, J = 17.5 Hz, J = 10.8 Hz), 5.76 (d, 1H, J = 17.5 Hz), 5.20 (d, 1H, J = 10.8 Hz), 2.96 (m, 2H). ^13^C NMR (DMSO-D6): 136.55, 135.08, 133.97, 133.89, 130.06, 125.91, 113.61, 35.62, 34.57. 31P NMR (DMSO-D6): 21.61. IR (KBr): *v* = 3560, 2755, 2320, 1630, 1510, 1410, 1270, 1230, 1100, 998 cm^−1^.

#### 2.2.3. VBCOQ

VBC was filtered over a short column of Alox B to remove radical inhibitors. The clear solution of VBC (30.5 g, 199 mmol) was dissolved in 100 mL chloroform and heated up to 60 °C. Then dimethyl octylamine (25 g, 158.9 mmol) was added slowly and the solution was kept at this temperature for 8 h. After cooling down to room temperature the product was precipitated in pentane, filtered and washed twice with pentane and dried in vacuum. VBCOQ was obtained as colorless powder (47.2 g, 96%). ^1^H NMR (CDCl_3_): 7.59 (2H), 7.41 (2H), 6.68 (dd, 1H, J = 17.6 Hz, J = 10.9 Hz), 5.79 (d, 1H, J = 17.6 Hz), 5.33 (d, 1H, J = 10.9 Hz), 4.91 (2H), 3.35 (m, 2H), 3.24 (6H), 1.76 (m, 2H), 1.25 (m 10H), 0.87 (t, 3H, J = 7.2 Hz), ^13^C NMR (CDCl_3_): 139.36, 135.48, 133.24, 126.76, 126.51, 115.81, 67.01, 62.84, 49.72, 31.62, 29.22, 29.02, 26.11, 22.59, 22.43, 13.88. IR: *v* = 2854, 1701, 1629, 1514, 1467, 1414, 1222, 863, 724 cm^−1^.

#### 2.2.4. VBCODQ

VBCODQ was prepared in the same manner as described in 2.2.3, using the following amounts: VBC (20 g, 131 mmol), dimethyl octadecylamine (31.2 g, 104.8 mmol), and chloroform (100 mL). VBCODQ was obtained as colorless powder (44.7 g, 94%). ^1^H NMR (CDCl_3_): 7.57 (4H), 7.41 (2H), 6.79 (dd, 1H, J = 17.6 Hz, J = 10.9 Hz), 5.09 (d, 1H, J = 17.6 Hz), 5.36 (d, 1H, J = 10.9 Hz), 4.57 (2H), 3.35 (m, 2H), 3.05 (6H), 1.89 (m, 2H), 1.30 (m 28H), 0.89 (3H), ^13^C NMR (CDCl_3_): 141.30, 137.04, 134.38, 128.19, 127.86, 116.48, 68.43, 65.73, 50.32, 30.76, 30.74, 30.72, 30.69, 30.60, 30.52, 30.43, 30.20, 27.43, 23.69, 23.66, 14.49. IR: *v* = 2917, 2849, 1629, 1515, 1469, 1414, 1223, 867, 719 cm^−1^.

### 2.3. Manufacture and Preparation of Ti Surface

Si wafers were cut in 15 mm × 15 mm squares and sputtered with a 100-nm thick Ti layer. Then, the discs were treated according to [77]; in detail, the discs were rinsed ultrasonically with 3 × 20 mL of hot toluene, followed by 3 × 20 mL of acetone, 3 × 20 mL of hot ethanol, and finally 4 × 20 mL of pure water. The disks were then placed in an oven at 120 °C overnight, to produce the hydroxylated titanium dioxide surfaces.

### 2.4. Silanisation of Ti Surface

The discs were pretreated with microwave plasma to generate hydroxyl moieties on the surface [79]. Then the discs were treated according to [77]; in detail, the discs were placed in a crystallizing dish with toluene (40 mL) and triethyl amine (1.4 mL). 4-(Chloromethyl)phenyltrichlorosilane (1.4 mL) was added and the solution was stirred overnight. The Ti discs were removed from the solution, rinsed with toluene and treated with ultrasonic treatment for 5 min in chloroform, acetone, and ethanol. Finally, the discs were dried at 70 °C for 20 min and kept in vacuum.

### 2.5. Polymerization

#### 2.5.1. Co-Polymerization VBCOQ with VBPOH

The polymerization was performed according to [80]. VBCOQ (8.0 g, 25.9 mmol) and *p*-vinyl benzyl phosphonic acid (1.3 g, 6.5 mmol) were dissolved in DMF/toluene (1:3) (100 mL) and heated up to 80 °C. AIBN (24 mg, 0.15 mmol) was added and the polymerization was kept for 48 h. The solvent was partially removed in vacuum, the polymer was precipitated in chilled diethyl ether and dried in vacuum to give a pale-yellow polymer (4.9 g).

#### 2.5.2. Co-Polymerization VBCODQ with VBPOH

Co-polymer of VBCODQ and VBPOH was prepared in the same manner as described in Section 2.5.1 using the following amounts: VBCODQ (11.7 g, 25.9 mmol), *p*-vinyl benzyl phosphonic acid (1.3 g, 6.5 mmol), AIBN (24 mg, 0.15 mmol), and DMF/toluene (1:3) (100 mL). A pale-yellow polymer (5.8 g) was obtained.

#### 2.5.3. ATRP with VBCOQ

The ATRP was performed according to [72,73]. VBCOQ (6.2 g, 20 mmol) was dissolved in DMF (40 mL) in a glass flask. Cu(I)Cl (40 mg, 0.40 mmol), Cu(II)Cl_2_ (22 mg, 0.16 mmol), and PMDETA (157 µL, 0.8 mmol) were added and the solution was stirred at room temperature for 40 min. Then the silane-modified Ti discs were placed around the stirring bar, the temperature was increased up to 70 °C and polymerization was done for 72 h. After cooling down, the discs were removed from the solution, rinsed two times with DMF and water, and treated with ultrasonic treatment two times for 10 min in DMF. Finally, the discs were dried at 40 °C for 2 h and kept in vacuum.

#### 2.5.4. ATRP with VBCODQ

The ATRP was prepared in the same manner as described in Section 2.5.1 using the following amounts: VBCODQ (9.0 g, 20 mmol), Cu(I)Cl (40 mg, 0.40 mmol), Cu(II)Cl_2_ (22 mg, 0.16 mmol), PMDETA (157 µL, 0.8 mmol), and DMF (40 mL).

### 2.6. Coating

General procedure: The Ti discs were pretreated with microwave treatment generating OH moieties [79]. The co-polymers were dissolved in methanol/dichloromethane (2:1) (40 mg/mL). Total of 75 µL of this solution was pipetted onto the Ti discs, and the Ti discs were dried at 40 °C for 24 h and treated with ultrasonic for 10 min in methanol/dichloromethane (2:1) for three times. Finally, the discs were dried at 40 °C and kept in vacuum.

### 2.7. Characterization

#### 2.7.1. Nuclear Magnetic Resonance (NMR) and Infrared (IR) Spectroscopy

^1^H- and ^13^C-NMR spectroscopy was used to characterize the chemical structures and compositions of the synthesized copolymer. The spectra were recorded on a Bruker DRX 400 spectrometer (Bruker BioSpin, Rheinstetten, Germany) at room temperature, using tetramethylsilane as an internal reference and CDCl_3_ as solvent.

IR spectra were recorded in the range from 4000 to 548 cm^−1^ by a Bio-Rad FTS 175 FT-IR Spectrometer with Golden Gate Reflection ATR (LOT Oriel, Darmstadt, Germany).

#### 2.7.2. XPS (X-ray Photoelectron Spectroscopy)

For the XPS measurements, the samples were examined using an AXIS Ultra DLD from Kratos Analytical Ltd. XPS (Manchester, Great Britain). The X-ray source emits monochromatic Al-Kα radiation (1486.8 eV). Overview spectra and detail spectra were recorded with power and pass energy of 150 W and 160 eV and 225 W and 20 eV, respectively.

#### 2.7.3. Cytotoxicity Assay

##### Cell Culture

Experiments to determine the cytotoxicity were performed with mouse-fibroblast cell line 3T3, obtained from the German collection of microorganisms and cell culture (DSMZ, Braunschweig, Germany). The cells were cultured in Dulbecco modified Eagle’s medium (DMEM) containing 10% fetal bovine serum and 50 U/mL penicillin and 50 µg/mL streptomycin at 37 °C and 5% CO_2_ atmosphere. In live/dead assay, MC3T3-E1 cells from mouse obtained from ATCC were cultured in α-medium containing 10% fetal bovine serum and 50 U/mL penicillin and 50 µg/mL streptomycin at 37 °C and 5% CO_2_ atmosphere. In vitro cytocompatibility determined by WST-1^®^ and live/dead assay. 

In the WST-1^®^ assay (Roche Diagnostics, Mannheim, Germany) eluates from the polymers were tested on cytotoxic components following manufacturer instruction. The test is used to evaluate the metabolic activities of cells by measuring the cleavage of WST-1 to formazan by mitochondrial dehydrogenases. Before starting the protocol, the polymers (three sample replicates of each kind of coating) were sterilized by ethanol (70%). The specimens were eluted by incubation in DMEM at 37 °C. According to the international standard DIN EN ISO 10993-5, 10993-12, respectively 1 mL of medium was applied per 0.2 g of material. Eluate samples were drawn after 1, 2, 3, 4, 7, and 14 days and the medium were completely renewed. The exposition of the eluates was done on pre-cultured 3T3 mouse fibroblasts on 96-well plate with a cell density about 8000 cells per well. From each sample replicate of the time four technical replicates were investigated. Pure cell culture medium applied to the cells defined the negative control. Wells without cells and thereby without any cellular enzymatic activity simulated the positive control. After 24 h incubation at 37 °C in a 5% CO_2_ atmosphere, cells were washed twice with sterile PBS (phosphate buffered saline) and incubated in fresh phenol red free medium with WST-1^®^ reagent (Roche Diagnostics, Mannheim, Germany) for about 1 h. WST-1^®^ conversion to formazan was measured at 450 nm. Relative dehydrogenase activities were calculated from the raw data by subtraction of the blank value (identical with the positive control) in the first step and relating to the negative control, the metabolic rate of uninfluenced cells, in a second step.

For the live/dead assay MC3T3-E1 cells were cultured on Ti discs for four days. After one and four days, two samples were rinsed with PBS and stained with 15 µg/mL fluorescein diacetate (FDA, Fluka, Taufkirchen, Germany), one fold GelRed^®^ (purchased 10,000 fold concentrated from VWR International, Darmstadt, Germany) in PBS. Living cells take up and deacetylate FDA to green fluorescent fluorescein. GelRed^®^ is able to permeate only damaged cell membranes and stains the nuclei of dead cells orange-red fluorescent. This staining procedure allows monitoring of changes in the ratio of live and dead cells in total cell populations.

#### 2.7.4. Bacterial Growth, Determination of Antibacterial Activity

*E. coli* DSM 1607, obtained from DSMZ, were cultured aerobically at 37 °C in CASO-Bouillon (Carl Roth, Karlsruhe, Germany) overnight before inoculation in experimental cultures. The Gram-negative bacterium *E. coli* is a standard model bacterium to determine antibacterial activity. The antibacterial activity was analyzed by measuring the ATP levels of bacterial suspension after exposition with titan specimens by using Bac-Titer-Glo-^TM^ Microbial Cell Viability Assay (Promega, Mannheim, Germany), a method for determining the number of viable microbial cells on quantitation of the ATP present. ATP, which is an indicator of living cells is detected by a luminescence reaction in which luciferin reacts with oxygen and ATP under luciferase catalysis to form oxyluciferin, AMP, and CO_2_, whereby light is emitted. The emitted luminescence signal is expressed as relative luminescence signal (RLU) and is proportional to the ATP level and thus also proportional to the number of living bacteria present. In short: a *E. coli* DSM 1607 suspension of 0.6 × 10^6^ CFU/mL was prepared in 0.9 % NaCl and 500 µL was applied to the various specimens in a 24-well plate. Untreated titan samples were used as reference samples. *E. coli* suspension without any sample were used as growth control. After 3 h incubation at 37 °C, 50 µL of bacterial exposure-suspension was added to Bac-Titer-Glo-reagent and after 5 min incubation luminescence measurements were performed using a TECAN Infinite M1000 Pro Plate Reader. RLU data from treated specimens were illustrated in comparison to Titan reference samples.

#### 2.7.5. Statistical Analysis

Data were expressed as mean with corresponding standard deviation for 12 replicates per specimens in WST-test and for 6 replicates in BTG-test. Data were analyzed with Origin and/or SigmaPlot. Analysis of variance (one-way ANOVA (BTG), Two-Way ANOVA (WST), Tukey post-hoc analysis) was used to assess differences between groups. A critical value of *p* < 0.05 was used to assess significance.

## 3. Results & Discussion

In this work, we present two new antibacterial polymers prepared by ATRP and compare them with polymers prepared via the classical route.

### 3.1. Monomer Synthesis

To prepare the polymers, we chose the route to synthesize the quaternary ammonium compounds first, which were polymerized in a second step. The advantage of that route is that no subsequent quaternization has to take place, where a complete quaternization rate of a polymer is often difficult to achieve. With our synthetic route, complete quaternization of the nitrogen is reached. The two quaternary ammonium salts VBCOQ and VBCODQ were synthesized in very good yields (96% (VBCOQ) and 94% (VBCODQ), respectively) from VBC and the corresponding dimethyl alkylamines (Figure 2).

Alkyl phosphonic acids or their esters are particularly suitable for bonding to titanium (oxide) surfaces due to their increased hydrolytic stability compared to silanes [77,81]. VBPOH was synthesized according to [78] (Figure 3). Nucleophilic substitution of VBC by sodium diethyl phosphite gave the diethyl phosphonate in 94% yield. Treatment of the diethyl phosphonate with trimethyl bromo silane gave the silyl ester intermediate. The removal of TMSBr must be carried out very carefully to avoid remaining traces of TMSBr which may form HBr during hydrolysis, catalyzing the polymerization of VBPOH. Careful hydrolysis and recrystallisation from acetonitrile gave the desired phosphonic acid in 55% yield.

### 3.2. Polymer Synthesis and Coating

#### 3.2.1. Co-Polymerization with VBPOH

The co-polymerizations of VBPOH with the two monomers proceeded according to standard conditions with AIBN as initiator [79]. The solubility of VBPOH is generally poor, so the polymerization solution remained turbid. The polymerization yields were only moderate at 53% (VBCOQ) and 44% (VBCODQ), respectively. The polymers were applied to the Ti surface by drop coating. Thermal treatment of the copolymer layer (approx. 40 °C) causes the phosphonic acid to form bonds with the surface titanium oxide layer. The formation of these titanium-oxygen-phosphorus bonds stabilizes the attachment of the copolymer layer to the titanium (oxide) layer [82]. In addition to phosphonic acids, their esters are also suitable for bonding to titanium [83], but they require high temperatures (approx. 120 °C). After 24 h, excess of non-bonded polymer was removed by ultrasound.

#### 3.2.2. ATRP

For the synthesis of polymer brushes on the titanium surface, a uniform and dense layer of initiator immobilized on the titanium surface is essential. Triethylamine (TEA)-catalyzed chlorosilanization of the Ti surface produced a stable initiator layer. The natural oxide layer of the titanium substrate was activated by a silane-coupling agent containing the ATRP initiator 4-(Chloromethyl) phenyltrichlorosilane (step 1, Figure 4) [72]. ATRP is controlled by an equilibrium between active and dormant chains, predominantly in the form of initiating alkyl halides/macromolecular species. The dormant chain reacts periodically with transition metal complexes in their lower oxidation state (here Cu(I)), which act as activators to form radicals (step 2). These radicals react with the double bond of the monomers (step 3). The deactivator (here (Cu(II)) reacts with the active chain in a reverse reaction to form the dormant chain and activator again (step 4). ATRP is a catalytic process and can be mediated by many redox-active transition metal complexes (CuI/L and X-CuII/L are the most commonly used transition metals). According to [72] the initial stage of ATRP needs, a sufficient concentration of the deactivating Cu(II) complex to quickly establish an equilibrium between the dormant and active chains. The Cu(II) complex can be added at the beginning of the reaction or can be obtained by the reaction of Cu(I) complex with the initiator [84]. In this work we choose the method of adding a Cu(II) complex (CuCl_2_) at the start of the reaction. The ratio of [monomer]/[CuCl]/[CuCl_2_]/[PMDETA] was controlled at 50:1:0.4:2. Hydrolytic stability of the resulting organic layer attached to the titanium surface is of crucial importance to the performance of the functionalized titanium substrate. The processes of immobilization of the ATRP initiator via the silane-coupling agent and the subsequent surface-initiated ATRP are shown schematically in Figure 4.

### 3.3. Characterization of the Polymer Layer

Determining the film thickness on very thin films is difficult, so we decided to use XPS to get an impression of the layer thickness. In the case of ATRP, no traces of Cu(I) or Cu(II) were found. Furthermore, the silicon of the ATRP initiator 4-(chloromethyl) phenyltrichlorosilane could be detected by XPS. A clear Ti 2p peak can be seen in all four samples (see Table 1 and Figure 5). Since the maximum information depth with XPS is 10 nm, a layer thickness significantly smaller than 10 nm can be assumed. Regarding to XPS, both coating variants result in a similar layer thickness. We used GDOES (glow discharge optical emission spectroscopy) as a second method for the determination of the layer thickness. With this method the results of XPS were confirmed.

### 3.4. Cytocompatibility Testing In Vitro

The cytocompatibility was tested using two different methods: the WST-1-test and the live/dead staining. In Figure 6, the results of the WST-1 assay are illustrated. In cytotoxicity testing, no decrease in dehydrogenase activities was observed in either the C8 or C18 QACs, but slightly increased values were recognized. Therefore, no evidence of cytotoxicity from released substances in either coating procedure were shown. However, all coatings indicated that cell viability was slightly higher than in Ti reference samples. But these findings were not supported by the live dead staining. The increasing as well the decrease of cell proliferation could result in an abnormal cell behavior as a reaction of the releasing substances from the coatings. The WST-1 assay is a sensitive, ready-to use standard method to assess the in vitro cytotoxicity by exposure to extracts. However, interference with the tetrazolium salt can influence the spectrophotometric measurement and result in misleading data [85,86].

At least, cells adhered directly onto the surface from the specimens should display cytotoxic effects of the materials themselves and whether high concentrations of cytotoxic components were formed or released at the surface. In live dead assay no increasing of proliferation in comparison to the untreated cell control and no difference in cell viability between the coating techniques or depending on the C-Side chain length was observed. Only a low number (less than 5 %) of orange-red fluorescent nuclei of dead cells were found independent from time but the cell density enhanced with time. In Figure 7 the results of live/dead staining are shown.

### 3.5. Antibacterial Testing

The Bac-Titer-Glo- and colony forming units test was conducted to assess the antibacterial activity of coating variants. The results are shown in Figure 8. The BTG data indicated that antibacterial effect against *E. coli* depends on the length of the side chain. VBCOQ (ATRP) and VBCOQ-co-VBPOH showed a lower number of viable bacteria after 3 h exposure compared to the VBCODQ (ATRP) and VBCODG-co-VBPOH. Further, it could be noted that there was no significant difference in antibacterial behavior between the coating variants. These findings were obtained with CFU analysis as well as with the BTG-tests.

### 3.6. Comparison of the Two Coating Variants

Differences between the two presumed coating types (polymer brushes vs. polymer film) were too small to detect. The reason for this may be that the layer thickness of less than 10 nm is too small. The fact that all coatings show no cytotoxicity in vitro, and moderate antibacterial activity, makes these coatings very interesting. Therefore, it is possible to decide depending on the type of sample, which coating variant appears to be the more favorable one. Drop coating is a very fast coating process which does not require high temperatures and is characterized by low material consumption. Drop coating does not allow complicated geometries to be coated and excess applied polymer must be removed. Furthermore, the applied polymer consists of at least two monomers, and in the case of VBPOH, poor yields have to be accepted in both, the synthesis of VBPOH and the polymerization.

Coating by ATRP makes it possible to coat even complicated specimen geometries. Polymerization can be carried out in a wide variety of solvents (including water) and a wide variety of metal salts (e.g., Ru, Fe, Ni, Mo, Mn, Os, Co [87]) can be used as catalysts. In the simplest case, homopolymerization takes place, but block polymerization is also possible. In the case of ATRP, rather long reaction times must be accepted and the specimen to be coated have to be immersed completely in the solution, which can easily lead to problems with more complicated geometries.

## 4. Conclusions

In the presented study new ultrathin polymeric coatings based on vinyl benzyl monomeric units bearing quaternary ammonium moieties have been prepared and tested with regard to their antimicrobial activity on titanium surfaces. The coatings were prepared by two different methods, on the one hand by direct ATRP surface polymerization of the corresponding QAC-containing monomers on plasma pre-activated titanium substrates, and for comparison, in a classical way by copolymerization of the QAC-containing monomers with a substrate-adhering vinylbenzyl phosphonate monomer and subsequent coating of the titanium substrates with the prepared copolymers by a drop coating procedure. The thickness of both coatings was investigated by XPS to be in the nanomolecular range below 10 nm. Furthermore, both coating types showed good substrate adhesion on smooth titanium surfaces and were found to be cytocompatible under in vitro conditions. Although the two coatings differ significantly in terms of their polymeric structure (brush-like structure of the ATRP-containing polymers versus planar polymeric chain structure of the polymers prepared by copolymerization in the classical way), polymer coatings showed a comparable moderate to good antimicrobial activity in killing bacteria on contact. In this context, differences in the antibacterial activity have been found by varying the chain length of the alkyl chain of the QAC. With *E. coli* used as test organisms, as carried out in our experiments, a higher antibacterial activity was determined using a QAC with an octyl (C8) chain compared to the use of an octadecyl (C18) chain. In further studies this finding has to be verified with other Gram-positive and Gram-negative bacterial test organisms and by further varying the length of the QAC alkyl chain length. With these further optimizations the developed coatings present a simple method to protect metallic devices against microbial contamination. Possible applications are the equipment of metallic devices used in the health care sector like door fittings or metallic bed frames; medical implants are also conceivable as potential applications. Furthermore, there is the possibility to equip the monomers with two long alkyl chains to generate an antiviral effect of the polymers [88].

## Figures and Tables

**Figure 1 nanomaterials-12-00614-f001:**
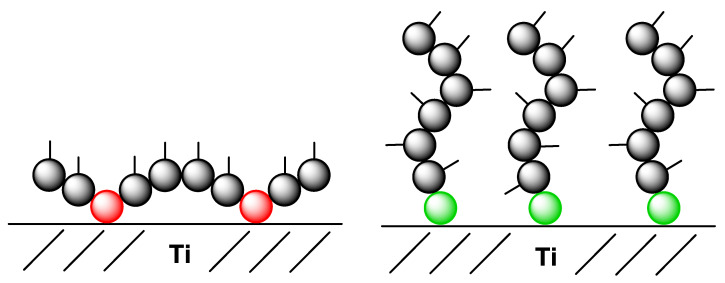
Different coating types. While the coating with the phosphonate co-polymer should lead to a planar-like-coating (**left**), the ATRP (atom transfer radical polymerization) should lead to polymer brushes (**right**). The phosphonates are marked in red and the anchoring-initiating molecules for the ATRP are marked in green. The grey balls represent the monomer units bearing the antibacterial quaternary ammonium group. The small lines represent the alkyl side chain.

**Figure 2 nanomaterials-12-00614-f002:**
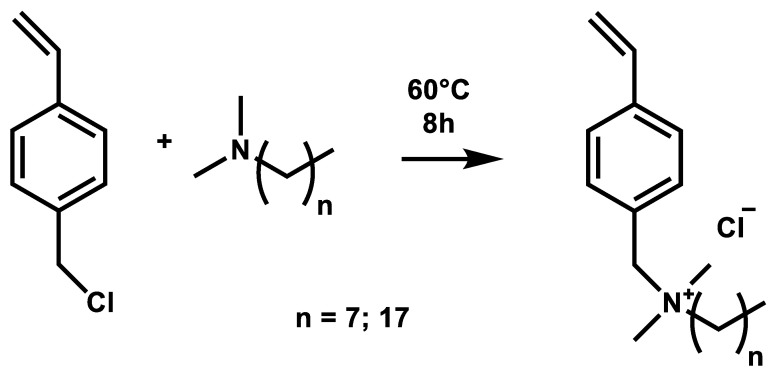
Synthesis of VBCOQ and VBCODQ.

**Figure 3 nanomaterials-12-00614-f003:**
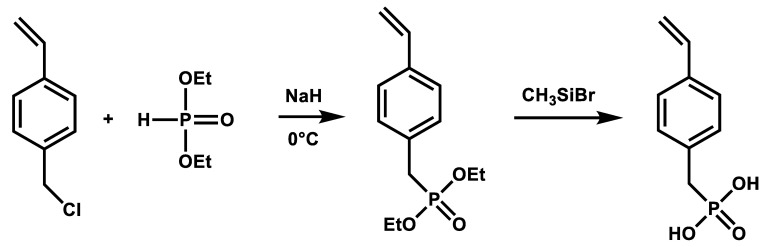
Synthesis of VBPOH.

**Figure 4 nanomaterials-12-00614-f004:**
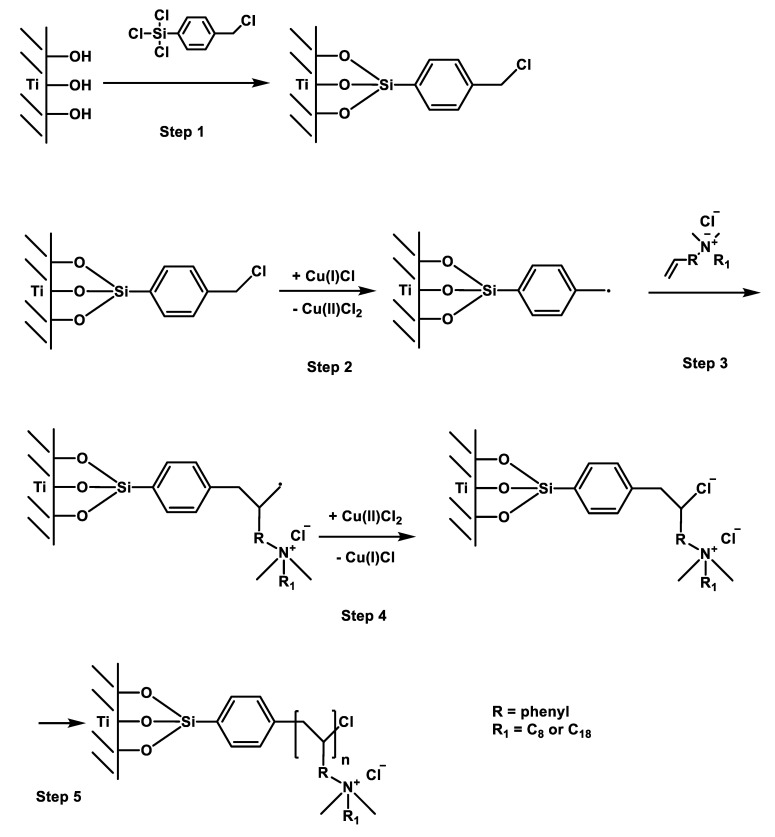
Scheme of an ATRP (atom transfer radical polymerization).

**Figure 5 nanomaterials-12-00614-f005:**
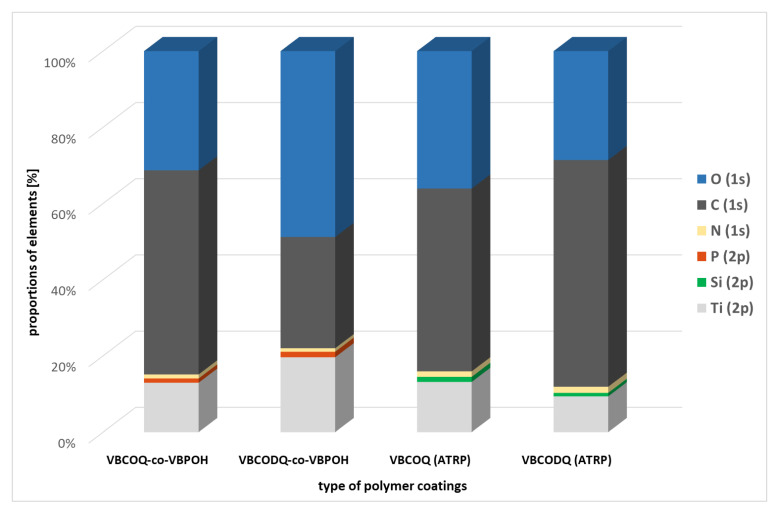
Graphical plot of the proportions (%) of the elements found with XPS (X-ray photoelectron *spectroscopy*).

**Figure 6 nanomaterials-12-00614-f006:**
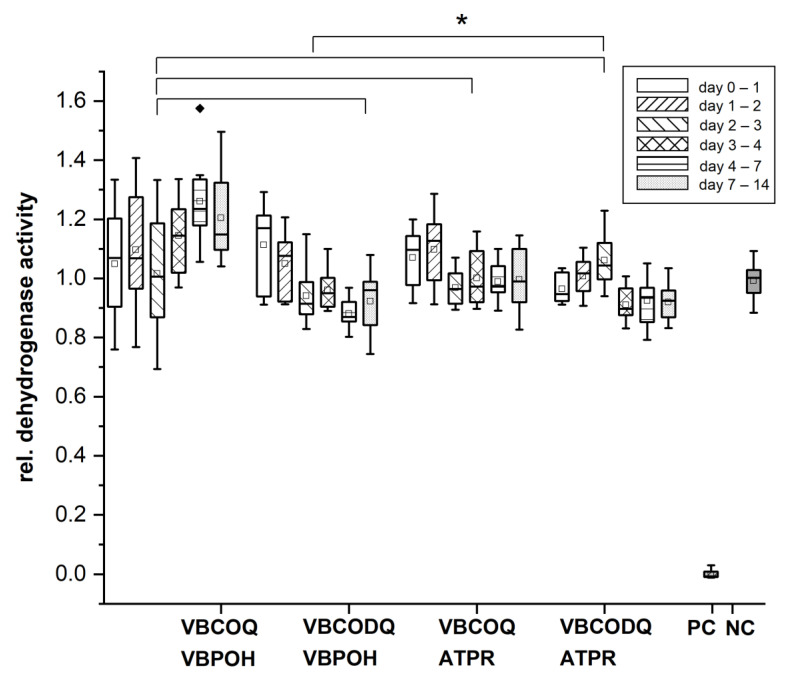
Results of a WST-1^®^ assay from consecutive eluates of both coating variants. The Box-whisker plots indicate the median in the center of the box, 25th and 75th percentile by the lower and upper margin of the box, 10th and 90th percentile by the whiskers and values outside this range. Boxes at a given time point represent values from elution media kept at the samples from the foregoing time point to the following time point. PC: positive control defining the highest level of enzyme inhibition realized by complete absence of cells, NC: negative control defined by uninfluenced cells. Two-way ANOVA, with Tukey multiple comparisons; *n* = 12, * *p* ≤ 0.05. Black diamond (◆) represent outlier.

**Figure 7 nanomaterials-12-00614-f007:**
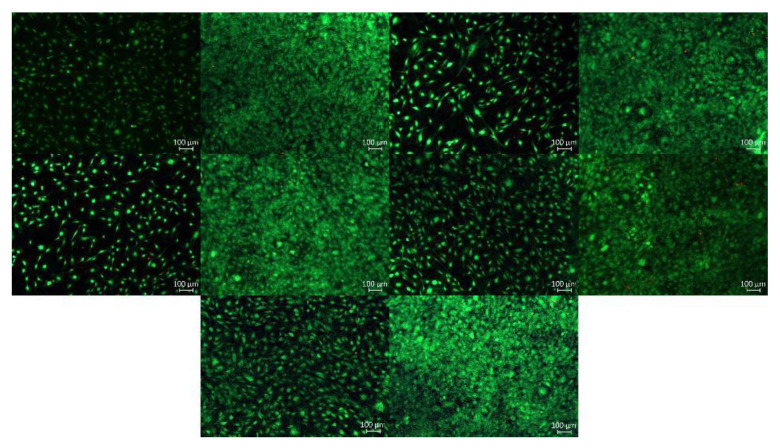
Results of the live/dead staining. Upper row (from **left** to **right**): VBCOQ (ATRP) day 1; VBCOQ (ATRP) day 4; VBCODQ (ATRP) day 1; VBCODQ (ATRP) day 4; middle row: VBCOQ-co-VBPOH day 1; VBCOQ-co-VBPOH day 4; VBCODQ-co-VBPOH day 1; VBCODQ-co-VBPOH day 4; bottom row: Ti reference day 1; Ti reference day4.

**Figure 8 nanomaterials-12-00614-f008:**
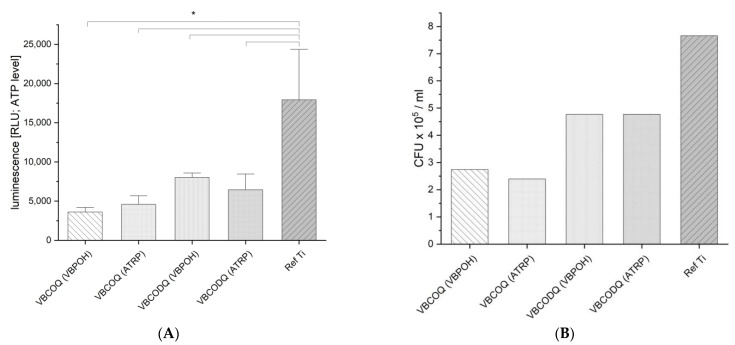
(**A**) Results of BTG-analysis (Bac-Titer-Glo™) as relative luminescence units, * *p* < 0.05 relative to titan reference (one-way ANOVA followed by a Tukey test; *n* = 8). Within the coating specimen no statistical significance was calculated. The data show that the VBCOQ is slightly more antibacterially active than the VBCODQ. A difference or trend in antibacterial activity in dependence of coating techniques ATRP and VBPOH could not be demonstrated. (**B**) Number of colony-forming units (CFU) determined from an aliquot of BTG exposure suspension. CFU confirmed the data of the BTG assay. The VBCOQ are slightly more antibacterial than the VBCODQ against *E. coli*. No difference results from the different coating techniques. Ref Ti = untreated titan reference.

**Table 1 nanomaterials-12-00614-t001:** Proportion [%] of elements found by XPS (X-ray photoelectron spectroscopy).

	VBCOQ-Co- VBPOH [%]	VBCODQ-Co- VBPOH [%]	VBCOQ (ATRP) [%]	VBCODQ (ATRP) [%]
O (1s)	31.21	48.67	36.08	28.59
C (1s)	53.44	29.14	47.98	59.52
N (1s)	1.06	0.92	1.44	1.61
P (2p)	1.11	1.45	–	–
Si (2p)	–	–	1.33	0.91
Ti (2p)	12.92	19.59	13.16	9.38

## Data Availability

We can provide required data upon request.

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
