# Peer review of "Comparable Studies on Nanoscale Antibacterial Polymer Coatings Based on Different Coating Procedures"

_nanomaterials, 2022, doi:10.3390/nano12040614_

Round 1

Reviewer 1 Report

 In this paper, the  antibacterial  activity  of  different  antibiotic  and  metal-free  thin  polymer  coatings  was  investigated. The films comprised quaternary ammonium compounds (QAC) based on vinyl benzyl  
chloride. First, two monomeric QAC with different alkyl chain lengths were prepared, which were then polymerized by two different polymerization processes and applied to Ti discs. On the one hand, a copolymerization with a phosphonate (VBPOH) was carried out and the Ti disks were coated via drop coating, with VBPOH providing covalent bonding to the Ti surface. On the other hand, a polymer layer was generated directly on the surface by Atom Transfer Radical Polymerization (ATRP). The coatings were characterized by GDOES and XPS. Cytocompatibility was evaluated by live/dead assay and WST- 1 assay, and antimicrobial activity was evaluated by BGT and CFU assay.  The experiment is well designed and the obtained reuslts are quite inetersting ,it can be publsihed after the minor revison. Comments: 

1,  Is the polymer layer stable on the surface?  How about the recycl ability ?

2, Is the thickness will afect the antimicrobial activity?

3, The recent paper are better to be mentioned in the introduciton part, Carbon, 2019, 155: 397-402; Carbon, 2018, 130: 775-781; Applied Catalysis B: Environmental, 2022, 301: 120826.  

Author Response

Dear reviewer,

thank you for you critical comments.

Here are our replies to your comments.

  1. Is the polymer layer stable on the surface? How about the recycl ability ?

The polymer layer is at least stable for 14 days. Further investigations of the stability of the polymer layer regarding humidity, temperature have to be done.

Recoating of samples is possible under the condition of pre-activation.

  1. Is the thickness will afect the antimicrobial activity?

We assume that thinner layers are more stable, since shearing off a thin layer is more difficult than shearing off a thick layer. Since this is a contact-active coating, we believe that the thickness of the polymer layer is no longer important above a certain thickness, since the deeper quaternary ammonium groups do not interact with the bacteria.

  1. The recent paper are better to be mentioned in the introduciton part,

Carbon, 2019, 155: 397-402; Carbon, 2018, 130: 775-781; Applied Catalysis B: Environmental, 2022, 301: 120826.

We added the mentioned papers to the reference list.

In the attached document you’ll find our respond to all reviewers.

With kind regards

Reviewer 2 Report

The subject of the manuscript “Comparable studies on nanoscale antibacterial polymer coatings based on different coating procedures” by Thorsten Laube et al. is focused on finding possible differences of the coating strategies in view of thickness or antibacterial activity. 

The manuscript can be accepted for publication after the authors will address all the raised queries (in the order they appear in the manuscript):  

  1. Almost 70% of the references used in the “Introduction” section date from at least one decade ago. The authors should therefore update the References list.
  2. “2.3. Preperation of the Ti discs” (page 4, line 153) – this Reviewer had some difficulties in understanding the exact nature of the used substrates. The title is about Ti discs but the authors begin the paragraph with Si wafers of (15×15) mm2. The authors should therefore reconsider this information.
  3. Who is the producer of the Ti discs?
  4. “PBS” (page 6, line 224) – an acronym should be described the first time it appears in the text.
  5. “Alkyl phosphonic …. to silanes [45; 47]” (page 6, lines 254 to 255) – this is not a result of this study. The authors should therefore move this paragraph to the “Discussion” section.
  6. “Hydrolytic stability …. titanium substrate” (page 8, lines 299 to 300) – this is not a result and the information should be therefore moved to the corresponding section.
  7. “Because determining ….. GDOES respectively” (page 9, lines 306 to 307) – this is not a result. Moreover, the authors already stated this info (subsection 2.7.2).
  8. “In GDOES ….. by a detector” (page 9, lines 308 to 315 and page 10, lines 316 to 317) – this is definitely not a result. All this paragraph should be therefore moved to the “Discussion” section.
  9. It is not clear why the authors used a non-calibrated machine when performing thickness investigations (qualitative GDOES tests). In this respect, the introduction of GDOES info is not justified. The authors should give a pertinent explanation for keeping this testing method as no results could be drawn.
  10. “Cell viability ….. dehydrogenase activity” (page 11, line 338) – this info was already mentioned in section 2.7.3.
  11. “Cells cultured in vitro directly at the materials” (page 11, line 352) – should be rephrased.
  12. “The BacTiter-Glo …….. bacteria present” (page 12, lines 364 to 369) – this is not a result. The authors should therefore move this paragraph to the corresponding section.
  13. Not all the results presented in Figure 9 have an error bar. Actually, the authors mention nothing about the statistics. As a consequence, this Reviewer considers the introduction of a section dedicated to statistics to be of paramount importance for this study. A discussion on the statistical relevance existing (or not) between the investigated samples should be also introduced.
  14. “It has been shown that with both coating variants the results are almost the same with regard to layer thickness” – this is not quite clear from the presented results.
  15. “By varying ….. more hydrophobic” (page 15, lines 415 to 418) – at least one reference should be introduced here.
  16. The “Conclusions” section should be rewritten taking into consideration the results obtained in this study. In addition, at least one possible application of the obtained results should be indicated.

Some minor recommendations follow:

- “need the bacteria come” (page 1, line 42) should read as “need the bacteria to come”;

- “stabilize” (page 7, line 277) should read “stabilizes”;

- “excess non-bonded” (page 7 lines 279 to 280) should read as “excess of non-bonded”;

- “at the begin of” (page 8, line 296) should read as “at the beginning of”;

- “found with XPS” (legend of Figure 6, page 11, line 331) should read “found by XPS measurements”;

- “the results are shown” (page 11, line 357) should read “the results of live/dead staining are shown”;

- “activities” (page 14, line 375) should read “activity”;

- “value” (page 14, line 387) should read “values”.

Author Response

Dear reviewer,

thank you for you critical comments.

Here are our replies to your comments.

  1. Almost 70% of the references used in the “Introduction” section date from at least one decade ago. The authors should therefore update the References list.

We updated the reference list.

  1. “2.3. Preperation of the Ti discs” (page 4, line 153) – this Reviewer had some difficulties in understanding the exact nature of the used substrates. The title is about Ti discs but the authors begin the paragraph with Si wafers of (15×15) mm2. The authors should therefore reconsider this information.

We clarified this section.

  1. Who is the producer of the Ti discs?

Informations about the producer of the Si wavers were added under 2.1 Materials.

  1. “PBS” (page 6, line 224) – an acronym should be described the first time it appears in the text.

Page 6 line 224 …sterile PBS… was amended to…with sterile phosphate buffer saline…  Page 6 line 230… phosphate buffer saline…was abridged to…PBS...

  1. “Alkyl phosphonic …. to silanes [45; 47]” (page 6, lines 254 to 255) – this is not a result of this study. The authors should therefore move this paragraph to the “Discussion” section.

We jointed the “result” and the “discussion” part into a single one.

  1. “Hydrolytic stability …. titanium substrate” (page 8, lines 299 to 300) – this is not a result and the information should be therefore moved to the corresponding section.

We jointed the “result” and the “discussion” part into a single one.

  1. “Because determining ….. GDOES respectively” (page 9, lines 306 to 307) – this is not a result. Moreover, the authors already stated this info (subsection 2.7.2).

We jointed the “result” and the “discussion” part into a single one.

  1. “In GDOES ….. by a detector” (page 9, lines 308 to 315 and page 10, lines 316 to 317) – this is definitely not a result. All this paragraph should be therefore moved to the “Discussion” section.

We jointed the “result” and the “discussion” part into a single one.

  1. It is not clear why the authors used a non-calibrated machine when performing thickness investigations (qualitative GDOES tests). In this respect, the introduction of GDOES info is not justified. The authors should give a pertinent explanation for keeping this testing method as no results could be drawn.

We restricted ourselves to the XPS results and deleted the GDOES investigations.

  1. “Cell viability ….. dehydrogenase activity” (page 11, line 338) – this info was already mentioned in section 2.7.3.

First sentence was removed.

  1. “Cells cultured in vitro directly at the materials” (page 11, line 352) – should be rephrased.

The Sentence was reworded to:

At least, cells adhered directly onto the surface from the specimens should display cytotoxic effects…

  1. “The BacTiter-Glo …….. bacteria present” (page 12, lines 364 to 369) – this is not a result. The authors should therefore move this paragraph to the corresponding section.

We jointed the “result” and the “discussion” part into a single one.

  1. Not all the results presented in Figure 9 have an error bar. Actually, the authors mention nothing about the statistics. As a consequence, this Reviewer considers the introduction of a section dedicated to statistics to be of paramount importance for this study. A discussion on the statistical relevance existing (or not) between the investigated samples should be also introduced.

Statistical analysis were done according the description in 2.7.5 (line 250 page 6). From CFU Value existed only two data per sample.

  1. “It has been shown that with both coating variants the results are almost the same with regard to layer thickness” – this is not quite clear from the presented results.

The following paragraph was added:

The BTG and CFU Test was conducted to assess the antibacterial activity of coating variants. The results are shown in Figure 9. The BTG data indicated that antibacterial effect against E.coli depends on the length of the side chain. VBCOQ (ATRP) and VBCOQ-co-VBPOH showed a lower number of viable bacteria after 3h exposure compared to the VBCODQ (ATRP) und VBCODG-co-VBPOH. Further, it could be noted that there was no significant difference in antibacterial behavior between the coating variants.

  1. “By varying ….. more hydrophobic” (page 15, lines 415 to 418) – at least one reference should be introduced here.

The section was deleted.

  1. The “Conclusions” section should be rewritten taking into consideration the results obtained in this study. In addition, at least one possible application of the obtained results should be indicated.

We revised the conclusion. A possible application was indicated.

Some minor recommendations follow:

- “need the bacteria come” (page 1, line 42) should read as “need the bacteria to come”;

The sentence was reworded to:

In non-leaching polymers it is indispensable for an antibacterial effect that the bacteria come very close to the surface, but the effect is not exhausted.

- “stabilize” (page 7, line 277) should read “stabilizes”;

- “excess non-bonded” (page 7 lines 279 to 280) should read as “excess of non-bonded”;

- “at the begin of” (page 8, line 296) should read as “at the beginning of”;

- “found with XPS” (legend of Figure 6, page 11, line 331) should read “found by XPS measurements”;

- “the results are shown” (page 11, line 357) should read “the results of live/dead staining are shown”;

- “activities” (page 14, line 375) should read “activity”;

- “value” (page 14, line 387) should read “values”.

The sentences were corrected as recommended.

In the attached document you’ll find our respond to all reviewers.

With kind regards

Reviewer 3 Report

Manuscript Number: nanomaterials-1553149

The manuscript entitled: Comparable studies on nanoscale antibacterial polymer coatings based on different coating procedures by Laube T et al., investigated the antimicrobial and cytotoxic activities of quaternary ammonium compounds, as polymer coatings, by means of different in vitro approaches.

Although the manuscript presents an interesting issue, unfortunately it is not suitable for publication on Nanomaterials.

In particular:

  • The introduction section is too generic (i.e. lines 22-24, 28-30, 33-35, 59-67) and the cited references too dated (i.e references number 1, 2, 4, 5, etc.).
  • The aim of work, at the end of the introduction section, is not exhaustively explained, so this lack creates a bit “of confusion” in the reader (line 92).
  • The materials and methods section presents many gaps in the methodologies used, particularly regarding sections 2.7.3 and 2.7.4.: the choice of the cell type and the bacterium used in the in vitro assays must be clarified.
  • The results and discussion sections should be rewritten taking into account logic on displaying results and appropriateness of discussion content; these two sections should be jointed in a single one.

Of note, taking into consideration the above-mentioned comment the abstract should be fully revised.

The cited references are a bit dated, it would be appropriate to add/replace with more recent works: 35 out of 54 references are before 2010.

The overall conclusion of the reviewer is that more details and clarifications are required to improve manuscript quality.

Author Response

Dear reviewer,

thank you for you critical comments.

Here are our replies to your comments.

  • The introduction section is too generic (i.e. lines 22-24, 28-30, 33-35, 59-67) and the cited references too dated (i.e references number 1, 2, 4, 5, etc.).

The references and the introduction were updated.

  • The aim of work, at the end of the introduction section, is not exhaustively explained, so this lack creates a bit “of confusion” in the reader (line 92).

The last paragraph of the introduction was rewritten.

  • The materials and methods section presents many gaps in the methodologies used, particularly regarding sections 2.7.3 and 2.7.4.: the choice of the cell type and the bacterium used in the in vitro assays must be clarified.

In line 227 page 6: The Gram-negative bacterium E.coli is a standard model bacterium to determine the antibacterial activity.

For determining the cytotoxicity 3T3 mouse fibroblast were used. This a standard and well stablished cell line for determining the biocompatibility, especially in WST-test. 3T3 fibroblast cells are heterogeny in morphology. That´s why we choose for testing the Live/dead staining the MC3T3 cells.

In line 200 page 5 the following issue was added:

Cell culture

Experiments to determine the cytotoxicity were performed with mouse-fibroblast cell line 3T3, obtained from the German collection of microorganisms and cell culture (DSMZ, Braunschweig, Germany). The cells were cultured in Dulbecco Modified Eagle's Medium (DMEM ) containing 10% fetal bovine serum and 50 U/ml Penicillin and 50 µg/ml Streptomycin at 37° C and 5% CO2 atmosphere. In live/dead assay MC3T3-E1 cells from mouse obtained from ATCC an cultured in α-Medium containing 10% fetal bovine serum and 50 U/ml Penicillin and 50 µg/ml Streptomycin at 37° C and 5% CO2 atmosphere.

  • The results and discussion sections should be rewritten taking into account logic on displaying results and appropriateness of discussion content; these two sections should be jointed in a single one.

We jointed these two sections into a single one.

Of note, taking into consideration the above-mentioned comment the abstract should be fully revised.

The abstract was fully revised.

The cited references are a bit dated, it would be appropriate to add/replace with more recent works: 35 out of 54 references are before 2010.

We updated the references.

In the attached document you’ll find our respond to all reviewers.

With kind regards

Reviewer 4 Report

Laube et al. have made a comparative study on nanoscale antibacterial polymer coatings based on different coating procedures. This is an interesting piece of work and possess adequate novelty which will be of interest to researchers in the field. After carefully reviewing this article, I suggest authors to revise their manuscript in accordance with the following comments before this article could be accepted for publication:

  1. Abstract – abbreviations such as GDOES and XPS should be expanded into full form.
  2. Keywords – please avoid abbreviations in keywords
  3. Introduction & rest of the manuscript – There are many 1, 2 or 3 sentence paragraphs which should be combined according to content into few large paragraphs. For example, in introduction, first three paragraphs should be combined as one, fourth to seventh paragraph as one, eighth and ninth as one. Likewise, it should be done in the rest of the manuscript as well.
  4. A schematic diagram should be included showing the various coating preparations described in the last paragraph of introduction along with the characterization and activity studies carried out in this study.
  5. The last sentence of introduction describing the objectives should be rewritten comprehensively for clarity and conciseness.
  6. Sections 2.2.3 and 2.2.4 – some initial few preparation steps are redundant and should be avoided.
  7. Sections 2.5.1 to 2.5.4 and 2.6 – reference citation should be included for these preparations.
  8. Section 2.7 (characterization) – Why there is not measurement of TGA/DSC analysis to determine the percentage polymer coating and their stability?
  9. L251 – what is RLU?
  10. Figure 5 – the x-axis and y-axis lines, labels and legends are not at all clear.
  11. Figures & Tables – all the abbreviations used should be described in the full form under the footnote for tables and in the notes following figure caption for figures. This is important as each figure & table should be independently understandable without reference to the text.
  12. Figure 6 – the x-axis and y-axis lines are missing. Also, there is not y-axis label.
  13. Conclusions – a collective summary of the key findings of this study is missing. Instead the authors have only provided the future perspective. It is suggested that a collective summary with key findings followed by future perspective should be provided under the conclusions.
  14. References – Some important references related to this work are not cited such as International Journal of Biological Macromolecules 2020, 161, 1484-1495; International Journal Nanomedicine 2014, 9, 5515-5531;
  15. Typographical errors should be double-checked throughout the manuscript.

Author Response

Dear reviewer,

thank you for you critical comments.

Here are our replies to your comments.

  1. Abstract – abbreviations such as GDOES and XPS should be expanded into full form.

We have explained the abbreviation in the text

  1. Keywords – please avoid abbreviations in keywords

Keywords are now without abbreviations.

  1. Introduction & rest of the manuscript – There are many 1, 2 or 3 sentence paragraphs which should be combined according to content into few large paragraphs. For example, in introduction, first three paragraphs should be combined as one, fourth to seventh paragraph as one, eighth and ninth as one. Likewise, it should be done in the rest of the manuscript as well.

We combined the paragraphs of the introduction and also for the rest of the manuscript.

  1. A schematic diagram should be included showing the various coating preparations described in the last paragraph of introduction along with the characterization and activity studies carried out in this study.

Figure 1 was redesigned and the aim of the work was revised.

  1. The last sentence of introduction describing the objectives should be rewritten comprehensively for clarity and conciseness.

The last paragraph of the introduction was rewritten.

  1. Sections 2.2.3 and 2.2.4 – some initial few preparation steps are redundant and should be avoided.

We deleted the redundant preparation descriptions in the mentioned paragraphs. Furthermore, we did the same in 2.5.

  1. Sections 2.5.1 to 2.5.4 and 2.6 – reference citation should be included for these preparations.

We added a reference for 2.5.1 and 2.5.4. For 2.5.3 and 2.5.4 there were already two reference stated. A reference for the microwave activation was added. We did the coating (2.6) without reference.

  1. Section 2.7 (characterization) – Why there is not measurement of TGA/DSC analysis to determine the percentage polymer coating and their stability?

The polymer amount on the samples is too low, so there is no detection of the weight loss in TGA.

  1. L251 – what is RLU?

In Page 6 Line 247ff …A more detailed description of BTG- and CFU- analysis was added, in  which the acronym RLU is specified.

  1. Figure 5 – the x-axis and y-axis lines, labels and legends are not at all clear.

We deleted Figure 5.

  1. Figures & Tables – all the abbreviations used should be described in the full form under the footnote for tables and in the notes following figure caption for figures. This is important as each figure & table should be independently understandable without reference to the text.

We described all abbreviations in the footnotes of the figures and tables.

  1. Figure 6 – the x-axis and y-axis lines are missing. Also, there is not y-axis label.

We added the labels for x- and y-axis.

  1. Conclusions – a collective summary of the key findings of this study is missing. Instead the authors have only provided the future perspective. It is suggested that a collective summary with key findings followed by future perspective should be provided under the conclusions.

We revised the conclusion.

  1. References – Some important references related to this work are not cited such as International Journal of Biological Macromolecules 2020, 161, 1484-1495;
    International Journal Nanomedicine 2014, 9, 5515-5531;

We added the mentioned papers to the reference list.

  1. Typographical errors should be double-checked throughout the manuscript.

We checked the manuscript.

In the attached document you’ll find our respond to all reviewers.

With kind regards

Round 2

Reviewer 2 Report

The manuscript can now be accepted for publication.

Author Response

Dear reviewer,

many thanks for your comments.

  • The manuscript can now be accepted for publication.

As there are no comments to reply, there is no need to answer.

In the attachment, you'll find our answers to the other reviewers.

With kind regards!

Reviewer 3 Report

Manuscript Number: nanomaterials-1553149

Even if the authors improved the overall quality of the manuscript, there is still an important issue related to the choice of cells (mouse-fibroblast cell line 3T3) and bacteria (E. coli DSM 1607) for in vitro assays. In fact, they should be chosen based on the clinical application of the biomaterial and, in the manuscript, it is not specified.

Hence: Why the authors decided to use fibroblasts? Again, why the authors used Escherichia coli?

The reported explanations:

“For determining the cytotoxicity 3T3 mouse fibroblast were used. This a standard and well stablished cell line for determining the biocompatibility, especially in WST-test. 3T3 fibroblast cells are heterogeny in morphology. That´s why we choose for testing the Live/dead staining the MC3T3 cells.” and “The Gram-negative bacterium E.coli is a standard model bacterium to determine antibacterial activity.” are not exhaustive.

Author Response

Dear reviewer,

thank you for your critical comment. Here are our answers:

  • Even if the authors improved the overall quality of the manuscript, there is still an important issue related to the choice of cells (mouse-fibroblast cell line 3T3) and bacteria (E. coli DSM 1607) for in vitro assays. In fact, they should be chosen based on the clinical application of the biomaterial and, in the manuscript, it is not specified.

Hence: Why the authors decided to use fibroblasts? Again, why the authors used Escherichia coli?

The reported explanations:

“For determining the cytotoxicity 3T3 mouse fibroblast were used. This a standard and well stablished cell line for determining the biocompatibility, especially in WST-test. 3T3 fibroblast cells are heterogeny in morphology. That´s why we choose for testing the Live/dead staining the MC3T3 cells.” and “The Gram-negative bacterium E.coli is a standard model bacterium to determine antibacterial activity.” are not exhaustive.

The fibroblasts were chosen because these cells represent an established model for the detection of cytotoxicity. (Lit) Fibroblasts and their functions play a pivotal role in the maintenance of structural stability of connective tissue architecture as well as in immune response. We choose the 3T3 fibroblasts as an appropriate cell type to get an impression about the cytotoxicity (proof of principle).

We addressed as possible applications metallic articles for use in medical or care facilities. Beyond this, the functionalization of implant surfaces is also included. For this background, the MC3T3 as progenitor cell lines in bone remodeling and the 3T3 cell line as most common cell type in connective tissue are an appropriate model system for demonstrating biocompatibility.

E.coli is a bacterium that is found on many surfaces. Especially in hospitals, E.coli is used as an indicator of hygiene in the clinical environment. If hygiene is poor, the bacterium can establish a variety of infections, especially in immunocompromised patients, including wound infections, urinary tract infections and even serious illnesses like sepsis.

We added the following sentence to the manuscript:

“…medical implants are also conceivable as potential applications.”

Lit (DOI: 10.1177/0748233718769806; PMID: 15624873, DOI: 10.1088/1748-6041/11/1/015021  )

In the attached file you'll find the answers to the other reviewers.
